# Stem Cell-Based Approaches for Spinal Cord Injury: The Promise of iPSCs

**DOI:** 10.3390/biology14030314

**Published:** 2025-03-20

**Authors:** Chih-Wei Zeng

**Affiliations:** Department of Neuroscience, University of Texas Southwestern Medical Center, Dallas, TX 75390, USA; chih-wei.zeng@utsouthwestern.edu

**Keywords:** induced pluripotent stem cells, spinal cord injury, regenerative medicine, cell-based therapies, neural differentiation, iPSC-derived cells

## Abstract

Spinal cord injuries can have life-changing consequences, often leading to permanent paralysis and loss of sensation. Currently, there are very few treatment options that can help the body repair itself after such injuries. However, stem cell research has opened new possibilities for recovery. In particular, a type of stem cell called induced pluripotent stem cells (iPSCs) offers hope for repairing damaged spinal cord tissue. iPSCs are unique because they can be created from a patient’s own cells and turned into different types of nerve cells that may help restore function. This article reviews the latest research on how iPSCs can be used to promote healing after spinal cord injuries, including how they can replace lost nerve cells, support existing cells, and reduce harmful inflammation. Scientists are also studying ways to make these treatments safer and more effective. While there are still challenges to overcome before these therapies can be widely used in hospitals, ongoing research is bringing us closer to making regenerative treatments a reality for people with spinal cord injuries. These advancements could greatly improve quality of life and independence for affected individuals in the future.

## 1. Introduction

Spinal cord injuries (SCIs) are life-changing events that affect millions of individuals worldwide, leading to motor and sensory impairments, chronic pain, and a reduced quality of life [1]. These injuries not only impact the lives of those directly affected but also place a significant burden on their families, caregivers, and society as a whole [2]. Currently, SCI remains challenging to treat due to the limited regenerative capacity of the central nervous system and the lack of effective curative treatments. Standard clinical management primarily focuses on minimizing secondary damage through surgical decompression, pharmacological interventions (e.g., corticosteroids), and extensive rehabilitation therapies, which only partially address the complex needs of SCI patients [3,4]. Thus, there is an urgent need for novel therapeutic strategies capable of promoting significant spinal cord regeneration and functional recovery.

As the medical community seeks effective treatments for SCIs, stem cell technology has emerged as a promising avenue for unlocking novel therapeutic approaches and deepening our understanding of the underlying disease mechanisms [5]. Several classes of stem cells have been explored for SCI treatment, including embryonic stem cells (ESCs), mesenchymal stem cells (MSCs), and induced pluripotent stem cells (iPSCs). ESCs exhibit high pluripotency but carry ethical concerns and risks of immune rejection [6]. MSCs possess immunomodulatory properties and low immunogenicity; however, their limited differentiation capacity restricts their regenerative potential for neural tissues [7,8]. Among these, iPSCs have gained considerable attention as particularly promising candidates in SCI research and treatment due to their ability to overcome these ethical and immunological challenges [9,10]. In this article, we will delve deeper into the exciting world of iPSCs and their applications in SCI models. One of the key strengths of iPSCs is their ability to be generated from a patient’s own somatic cells, which allows for the creation of patient-specific cell lines [11]. These personalized cell lines can then be used to model SCIs, study disease mechanisms, and develop tailored treatments. Furthermore, iPSCs can be differentiated into various cell types relevant to SCIs, such as neurons, astrocytes, oligodendrocytes, and microglia [12,13,14,15]. This enables researchers to investigate the complex interplay between different cell types in the context of SCIs and develop cell replacement therapies aimed at promoting functional recovery.

iPSCs have also proven valuable in drug discovery and screening, as they allow researchers to test the safety and efficacy of potential treatments in a controlled laboratory setting before advancing to clinical trials [16]. By providing a more accurate representation of the human spinal cord and its response to injury, iPSC-derived models can improve the likelihood of identifying effective therapies and reduce reliance on animal models. Despite the tremendous potential of iPSCs in SCI research, challenges remain. Ensuring the safety of iPSC-based therapies is a major concern, as the reprogramming process can introduce genetic mutations or epigenetic changes that could lead to tumorigenesis [17,18,19,20]. Additionally, optimizing the survival, integration, and functionality of transplanted cells in the host spinal cord is crucial for the success of cell-based therapies [21]. Nevertheless, iPSCs offer significant advantages over other stem cells due to their ethical acceptability, reduced immunological rejection risks through autologous transplantation, and versatile differentiation capacity, positioning them as highly promising candidates for advancing regenerative medicine and SCI treatment strategies [9,10].

As we continue to explore and harness the potential of iPSCs in SCI research and treatment, there is hope that these groundbreaking cells will contribute significantly to enhancing the lives of those affected by SCI. By driving innovation in research methodologies and treatment strategies, iPSCs may ultimately pave the way for a new era of SCI understanding, therapy, and, perhaps one day, even a cure.

## 2. iPSCs: A Breakthrough in SCI Research

iPSCs, first generated by Kazutoshi Takahashi in collaboration with Shinya Yamanaka’s laboratory in 2006, have revolutionized the field of stem cell research [9]. These adult cells are reprogrammed into an embryonic-like state, enabling them to differentiate into various cell types [22]. Their derivation from a patient’s own cells circumvents issues related to immune rejection and ethical concerns associated with embryonic stem cells, making them an attractive option for research and potential therapies [23,24]. In the context of SCI research, iPSCs have emerged as a powerful tool with the potential to transform our understanding and treatment of these debilitating injuries. 

SCI typically results from traumatic events such as vehicle accidents, falls, sports injuries, or violence. Non-traumatic causes such as infections, tumors, or degenerative diseases can also lead to SCI. The pathology of SCI involves primary mechanical injury causing immediate tissue disruption, followed by secondary injury mechanisms, including inflammation, oxidative stress, neuronal apoptosis, and glial scar formation, all of which complicate recovery and treatment strategies [4].

Some key applications of iPSCs in SCI research include the following.

### 2.1. Patient-Specific Disease Modeling

By generating iPSCs from individuals with SCI, researchers can create personalized in vitro models that accurately represent the patient’s genetic background and injury characteristics [25,26]. This enables scientists to study the underlying molecular mechanisms of SCIs, investigate the impact of genetic factors on injury susceptibility and recovery, and explore the efficacy of potential treatments in a patient-specific manner.

### 2.2. Cellular Reprogramming and Differentiation

iPSCs can be differentiated into a wide range of neural cell types relevant to SCIs, such as neurons, astrocytes, and oligodendrocytes (Figure 1). Typically, iPSCs can be derived either from a patient’s own cells (autologous) or from healthy donors (allogeneic). While autologous iPSCs significantly reduce immune rejection risks, their generation and validation are time-consuming, costly, and labor-intensive. In contrast, allogeneic iPSC-derived therapies, especially those sourced from HLA-matched or universal donor cell banks, offer readily available standardized cell products suitable for broader clinical applications, allowing for more immediate therapeutic use [9]. The differentiation and survival of iPSCs are strongly influenced by their cellular niche or microenvironment, including signaling molecules, extracellular matrix components, and inflammatory mediators present after SCI [9]. This feature allows researchers to study the complex interplay between various cell types in the context of SCI and develop cell replacement therapies aimed at promoting functional recovery. Furthermore, the development of sophisticated differentiation protocols has facilitated the generation of subtype-specific neural cells [27,28], allowing for more precise investigations into the distinct contributions of various cell types in both the injury and recovery processes. By better understanding these cellular interactions and the mechanisms underlying SCI, researchers can identify potential therapeutic targets and design more effective strategies to enhance tissue repair and functional regeneration. 

### 2.3. Drug Discovery and Screening

iPSC-derived SCI models enable researchers to test the safety and efficacy of potential treatments in a controlled laboratory setting before advancing to clinical trials [29]. These models offer a more accurate representation of the human spinal cord and its response to injury, thereby enhancing the chances of discovering successful therapies. Additionally, the use of iPSC-derived models can help reduce the reliance on animal models in SCI research [30]. By closely mimicking the human condition, iPSC-based models allow for better translation of experimental findings to real-world clinical applications, ultimately increasing the probability of developing effective therapies for SCI.

### 2.4. Precision Medicine and Personalized Therapies

The use of patient-specific iPSCs paves the way for the development of tailored treatments that take into account an individual’s unique genetic and molecular profile [31]. This approach has the potential to improve the effectiveness of therapies and minimize adverse effects, ultimately leading to better outcomes for those affected by SCI [32]. By generating iPSCs from a patient’s own cells, researchers can study the precise mechanisms underlying their specific injury and response to treatment. This, in turn, allows for the identification of targeted, patient-specific interventions that address the underlying causes of the injury and promote more efficient recovery. Furthermore, personalized therapies have the potential to minimize the risk of immune rejection, as the transplanted cells are derived from the patient’s own body, thus avoiding complications related to immune compatibility [33]. The integration of iPSCs into precision medicine approaches holds great promise for the development of innovative, effective, and safe therapies that can significantly improve the quality of life for individuals with SCI [34,35], paving the way for a new era of personalized medicine in the field of SCI treatment and research.

### 2.5. Concerns Regarding Culture Media for iPSC Induction

Different culture medium formulations required for generating and differentiating iPSCs present specific concerns regarding clinical translation. The choice of medium can significantly influence cell quality, stability, differentiation efficiency, and potential safety risks. To ensure clinical applicability, researchers prioritize defined, feeder-free media and incorporate optimized combinations of growth factors and small molecules that reduce genetic and epigenetic variability and tumorigenic risks associated with iPSC therapies [9].

Despite these promising applications, challenges remain in the field of iPSC-based SCI research. Ensuring the safety of iPSC-derived therapies, optimizing the survival and integration of transplanted cells, and understanding the complex host environment of the injured spinal cord are all critical areas that require further investigation. However, with continued research and technological advancements, iPSCs hold the potential to revolutionize our understanding and treatment of SCIs, bringing hope to those affected by these life-altering events.

## 3. Cell-Based Therapies for SCI

iPSCs have emerged as a promising tool for developing cell-based therapies to treat SCI. As highlighted in the study, iPSCs can be differentiated into various cell types that play crucial roles in SCI treatment, including neural progenitor cells, oligodendrocyte progenitor cells, astrocytes, and microglia (Figure 1). The transplantation of these cells into injured spinal cords in animal models has shown promising results, with several studies reporting improvements in functional recovery [36,37]. Here, we discuss some of the key aspects of iPSC-based cell therapies for SCI.

### 3.1. Neural Progenitor Cells

iPSC-derived neural progenitor cells (NPCs) have the potential to differentiate into various neuronal and glial cell types [38,39], making them a promising candidate for cell replacement therapies in SCI. Transplantation of NPCs into injured spinal cords has shown beneficial effects, such as axonal regeneration, remyelination, and synapse formation [40,41]. These processes aid in restoring neural circuitry and enhancing motor and sensory function in animal models [42]. For instance, Lu et al. (2012) demonstrated that transplantation of human iPSC-derived NPCs into a primate model of SCI resulted in significant motor function improvement, as assessed by behavioral scoring systems [43]. Nori et al. (2011) also demonstrated that transplantation of human iPSC-derived neural progenitor cells significantly enhanced motor function recovery in mouse models of spinal cord injury, as assessed by behavioral tests [44]. In addition to their direct regenerative potential, NPCs also exhibit paracrine effects, such as secreting neurotrophic factors and modulating the local immune response, which play significant roles in promoting tissue repair and reducing inflammation in the injured spinal cord [45]. For example, NPCs have been found to secrete brain-derived neurotrophic factor (BDNF) and glial cell-derived neurotrophic factor (GDNF), which support neuronal survival and axonal regeneration [46]. Additionally, NPCs can also release anti-inflammatory cytokines like interleukin-10 (IL-10), contributing to the attenuation of the local inflammatory response [47]. Moreover, NPCs exert paracrine effects, like releasing neurotrophic factors (e.g., CNTF and NT-3) [48], which can aid in tissue repair and inflammation reduction in the injured spinal cord. These multifaceted properties not only address cell loss but also facilitate a more supportive environment for endogenous repair mechanisms. Consequently, iPSC-derived NPCs offer great potential for developing effective cell-based therapies aimed at improving functional recovery following SCI. Future research should focus on optimizing NPC transplantation protocols, including timing, dosage, and delivery methods, to maximize the therapeutic potential of these cells in SCI treatment.

### 3.2. Oligodendrocyte Progenitor Cells

Oligodendrocytes play a crucial role in forming myelin sheaths around axons, ensuring efficient nerve conduction and overall neuronal function [49,50]. In the wake of SCI, demyelination often affects surviving axons, leading to functional impairments and exacerbation of damage. iPSC-derived oligodendrocyte progenitor cells (OPCs) present a promising therapeutic approach to promote remyelination and restore neuronal function post-SCI [51,52]. A Phase I clinical trial investigated the safety of human embryonic stem cell-derived OPCs (GRNOPC1) in patients with subacute thoracic SCI. The study demonstrated that GRNOPC1 could be safely delivered to the injury site, with patients showing no serious adverse events and potential signs of reduced spinal cord tissue deterioration [53]. The transplantation of iPSC-derived OPCs into injured spinal cords has been shown to facilitate oligodendrocyte maturation, resulting in new myelin sheath formation around demyelinated axons [14]. This remyelination process can boost nerve conduction and consequently enhance functional recovery. Studies have indicated that OPC transplantation leads to significant locomotor recovery in animal models of SCI [54,55]. In addition to their remyelination capabilities, iPSC-derived OPCs have been observed to secrete neurotrophic factors, including IGF-1 [54], and modulate the inflammatory environment at the injury site, further enhancing their therapeutic potential. OPCs with elevated expression of PDGF-AA have been linked to enhanced myelination and tissue repair following SCI, which contribute to the restoration of neurological function [56]. By addressing both demyelination and the local injury environment, OPC transplantation offers a comprehensive approach to SCI treatment, raising the likelihood of successful functional recovery.

### 3.3. Astrocytes

Astrocytes are essential glial cells that play a critical role in maintaining the homeostasis and structural integrity of the central nervous system. They contribute to a variety of functions, such as providing metabolic support to neurons, modulating synaptic transmission, and upholding the blood–brain barrier [57]. In a preclinical study, human iPSC-derived astrocytes transplanted into a rodent model of SCI promoted neuronal survival and axonal regeneration, resulting in significant improvements in motor function, as measured by the Basso, Beattie, and Bresnahan (BBB) locomotor rating scale [58]. iPSC-derived astrocytes have demonstrated the capacity to support the survival and function of neurons and oligodendrocytes in both in vitro and in vivo settings [59,60]. Introducing iPSC-derived astrocytes into injured spinal cords offers a promising therapeutic approach in SCI. Once administered to the injury site, these astrocytes can help regulate the inflammatory response by releasing anti-inflammatory factors, such as IL-10 and TGF-β, and mitigating the activation of harmful microglial cells [61,62]. This process reduces secondary damage and fosters a more favorable environment for tissue repair. In addition to their immunomodulatory properties, iPSC-derived astrocytes can promote tissue repair by secreting neurotrophic factors, such as CNTF and GDNF, and providing structural support to regenerating axons, ultimately improving functional outcomes in animal models [63,64]. Furthermore, astrocytes have been found to play a role in regulating the formation of glial scars, physical and biochemical barriers that can impede axonal regeneration [65,66]. Modulating astrocyte behavior and interactions with glial scars, for example, by using chondroitinase ABC to degrade inhibitory chondroitin sulfate proteoglycans, may potentially enhance the regenerative capacity of injured spinal cords [67].

### 3.4. Microglia

Microglia, as the primary immune cells of the central nervous system (CNS), play a pivotal role in maintaining homeostasis and responding to injury or infection [68,69]. In SCI scenarios, activated microglia can display both pro-inflammatory and anti-inflammatory properties, which can influence the extent of spinal cord damage [70,71]. While direct transplantation of iPSC-derived microglia into SCI models is still under investigation, studies have utilized these cells to understand microglial responses in neuroinflammation, which may inform future therapeutic strategies targeting microglial modulation to enhance recovery post-SCI. iPSC-derived microglia serve as a valuable tool for examining these immune cells’ complex roles and interactions with other cell types in the injured spinal cord and their contribution to SCI pathology [72]. Utilizing patient-specific iPSCs to generate microglia-like cells enables researchers to investigate the molecular mechanisms and signaling pathways involved in microglial activation, polarization, and function in a personalized and physiologically relevant context [73,74]. For example, iPSC-derived microglia can be employed to study the role of the TREM2 receptor in microglial activation [75,76] and its potential contribution to SCI progression. Another example includes examining the involvement of the CX3CR1/CX3CL1 signaling pathway in microglial–neuronal communication and the regulation of microglial neuroprotective functions [77]. By developing a comprehensive understanding of microglia’s role in SCI, it becomes possible to create targeted therapeutic interventions that modulate microglial activity in a controlled manner. Such interventions may involve promoting the tissue-repairing properties of microglia while minimizing their pro-inflammatory effects, which can worsen tissue damage [78,79]. Exploring combinatorial treatment strategies that incorporate iPSC-derived microglia alongside other regenerative medicine approaches, rehabilitation, or pharmacological interventions may further enhance patient outcomes [80]. Ultimately, leveraging iPSC-derived microglia to develop and optimize these interventions will contribute to more effective and comprehensive treatment strategies for individuals affected by SCI.

### 3.5. Combination Therapies

Considering the intricate nature of SCI pathology, which involves multiple cell types and molecular pathways, there is a growing interest in developing combination therapies that integrate different iPSC-derived cell types or incorporate additional therapeutic strategies. For example, combining iPSC-derived NPCs with biomaterial scaffolds in SCI models has provided structural support and promoted cell survival, leading to enhanced motor function recovery compared to cell transplantation alone [81]. These multifaceted approaches aim to tackle the numerous challenges associated with SCI repair and regeneration, ultimately fostering synergistic effects that can lead to enhanced functional recovery. By combining iPSC-derived cell types, such as NPCs, OPCs, astrocytes, and microglia, researchers can create a more comprehensive therapeutic strategy that addresses multiple aspects of the injury simultaneously [82,83]. This approach has the potential to amplify the overall efficacy of cell-based therapies by promoting axonal regeneration, remyelination, and modulation of the inflammatory response. Additionally, combining iPSC-based therapies with biomaterial scaffolds and growth factors can further optimize the therapeutic outcome by providing physical support, enhancing cell survival, and delivering crucial signaling molecules to the injury site [84,85].

In addition to combining different cell types, researchers are also exploring the use of other therapeutic strategies in conjunction with iPSC-based therapies. These may include the following.

#### 3.5.1. Biomaterial Scaffolds

Biomaterial scaffolds play a crucial role in iPSC-based therapies for SCI, providing a supportive structure for transplanted cells while encouraging their survival, migration, and integration into the host tissue [85,86]. These scaffolds can be categorized into natural, synthetic, and hybrid biomaterials. Natural biomaterials such as collagen, chitosan, alginate, and fibrin offer excellent biocompatibility and resemble native extracellular matrix (ECM) structures, supporting cellular adhesion and differentiation. Synthetic biomaterials, including poly(lactic-co-glycolic acid) (PLGA), polyethylene glycol (PEG), and poly(caprolactone) (PCL), provide precise control over mechanical properties, degradation rates, and structural characteristics [87,88]. Hybrid scaffolds combine properties of both natural and synthetic materials, leveraging their respective advantages for optimal tissue integration and regeneration [89]. They can be tailored to resemble the extracellular matrix, delivering physical and biochemical cues that promote cell growth, differentiation, and tissue regeneration [90,91,92]. Moreover, immunomodulatory biomaterials, such as chitosan-based or hyaluronic acid-based scaffolds, can modulate the local immune response, reducing inflammation and improving the survival and integration of transplanted iPSCs [87]. Furthermore, biomaterial scaffolds can be engineered to release growth factors, such as nerve growth factor (NGF), BDNF, or GDNF, or other bioactive molecules like anti-inflammatory agents and extracellular matrix components [93,94,95]. These factors can enhance the therapeutic potential of transplanted cells, modulate the inflammatory response, and stimulate axonal regeneration [96,97]. By incorporating biomaterial scaffolds into iPSC-based therapies, researchers can create a more hospitable environment for cell survival and function, ultimately improving the chances of successful tissue repair and functional recovery following SCI.

#### 3.5.2. Growth Factors

Growth factors are crucial in iPSC-based therapies for SCI, providing essential molecules that encourage cell survival, differentiation, and tissue repair. Several growth factors with therapeutic potential include BDNF, promoting neuronal survival and growth [98]; vascular endothelial growth factor (VEGF), stimulating angiogenesis and improving blood supply to the injured region [99]; and NT-3, aiding in the survival and development of various neuronal populations [100]. Growth factors function through specific signaling pathways: BDNF primarily acts through the TrkB receptor to activate signaling cascades promoting neuronal survival, differentiation, and synaptic plasticity [101]. NGF exerts its effects mainly through the TrkA receptor, enhancing neurite outgrowth, survival, and differentiation of sensory and sympathetic neurons [102]. Similarly, fibroblast growth factors (FGFs) bind to FGF receptors, initiating signaling pathways that support cell proliferation, differentiation, and neurogenesis [103]. Integrating these growth factors into iPSC-based treatments can amplify the regenerative abilities of transplanted cells and establish a more conducive environment for tissue repair. Various strategies can be employed to deliver growth factors, such as genetically modifying transplanted cells to overexpress the desired factors [104], incorporating them into biomaterial scaffolds [105], or utilizing controlled-release systems for direct administration [106]. By fine-tuning the delivery methods and dosages of these essential signaling molecules, researchers can increase the efficacy of iPSC-based therapies and foster functional recovery after SCI.

#### 3.5.3. Electrical Stimulation

Electrical stimulation is an emerging approach to improve the outcomes of iPSC-based therapies for SCI [21,107]. By applying controlled electrical currents to the injured spinal cord, it is possible to enhance neural activity, promote axonal regeneration, and facilitate functional recovery [108]. Electrical stimulation can be applied either directly to the spinal cord or indirectly through the surrounding muscles, depending on the specific goals and requirements of the therapy. When combined with iPSC-based treatments, electrical stimulation can have synergistic effects, such as improving the survival and integration of transplanted cells [109], modulating the inflammatory response [110], and promoting the formation of new synaptic connections [111]. Moreover, electrical stimulation can help to restore the activity of neural circuits and enhance neuroplasticity, which is crucial for the recovery of motor and sensory functions following SCI [112,113]. Ongoing research is focused on refining the techniques and technologies used for electrical stimulation and exploring the potential benefits of combining this approach with iPSC-based cell therapies to maximize the functional recovery of SCI patients.

#### 3.5.4. Therapeutic Potential of iPSC-Derived Exosomes

Exosomes derived from iPSCs have recently emerged as promising therapeutic tools for spinal cord injury. These extracellular vesicles can modulate the injury microenvironment through their anti-inflammatory, neuroprotective, and regenerative properties [114,115]. Specifically, iPSC-derived exosomes can reduce inflammation by suppressing pro-inflammatory cytokines and promoting polarization of microglia towards a neuroprotective phenotype [116]. Furthermore, they have been shown to facilitate neural regeneration by delivering bioactive molecules such as microRNAs, proteins, and growth factors directly to target cells, thereby promoting axonal regrowth, neuronal survival, and functional recovery [117]. Due to their nano-scale size, low immunogenicity, and ability to cross biological barriers, iPSC-derived exosomes represent an attractive, cell-free therapeutic approach that could complement or even replace traditional cell transplantation strategies [115,118]. Continued investigation into the optimal isolation methods, dosing strategies, and delivery routes will further enhance their clinical applicability and potential in SCI therapies.

These combination therapies represent a promising approach to addressing the multifaceted nature of SCI pathology. By targeting multiple aspects of the injury and harnessing the synergistic effects of various therapeutic strategies, researchers hope to achieve more significant improvements in functional recovery, ultimately benefiting those affected by SCI. Further research is needed to optimize these combination therapies and evaluate their safety and efficacy in preclinical and clinical settings. Overall, despite the promising results observed in preclinical studies, several challenges remain for the translation of iPSC-based cell therapies to the clinic (Table 1). These include ensuring the safety and efficacy of the transplanted cells, optimizing cell delivery methods, and understanding the long-term effects of cell transplantation on the host spinal cord. As research continues to advance in this field, iPSC-based cell therapies hold great promise for the development of novel, effective treatments for SCIs, potentially transforming the lives of those affected by these devastating conditions.

## 4. Clinical Research: Current Status and Prospects

The current status of clinical research on iPSC-derived cell therapies for SCI is marked by an increasing number of trials to evaluate their safeties and feasibilities [119,120]. These trials are predominantly involved in transplanting iPSC-derived neural progenitor cells or oligodendrocyte progenitor cells, with a focus on safety parameters such as adverse events, tumorgenicity, and immunological responses [121,122,123]. While early trials’ results demonstrated that they are promising, it is crucial to conduct further large-scale studies to establish the therapies’ effectiveness.

Personalized medicine approaches using patient-specific iPSCs have been highlighted in several preclinical and clinical studies. For example, Nori et al. (2015) demonstrated the successful transplantation of patient-specific iPSC-derived neural stem cells into non-human primate SCI models, resulting in improved motor function and reduced immune rejection [124]. Similarly, clinical trials such as those by Nagoshi and Okano (2017) have shown the feasibility of using autologous iPSC-derived neural progenitor cells for patients with SCI, with preliminary evidence of safety and modest functional improvements [40]. Furthermore, gene-editing techniques like CRISPR-Cas9 combined with patient-derived iPSCs have allowed researchers to correct genetic mutations prior to transplantation, further enhancing the therapeutic potential and customization of SCI treatments [125]. Despite these advances, challenges remain, including the extensive time, resources, and technical expertise required for patient-specific iPSC generation and validation. Comparative analyses of autologous versus allogeneic approaches continue to be critical in determining optimal strategies for clinical translation.

Looking towards future prospects, several challenges must be addressed to advance the clinical translation of iPSC-derived cell therapies (additional details are also discussed in the following section). First, establishing standardized protocols for differentiating, characterizing, and ensuring the quality of iPSC-derived cells intended for transplantation is essential [126,127]. Second, determining the optimal timing, dosage, and delivery methods for cell transplantation will maximize their therapeutic potential [128,129]. Third, the development of biomarkers and imaging techniques is crucial for monitoring transplanted cells’ survival, integration, and functional impact in vivo [130]. Additionally, the exploration of combinatorial strategies, including iPSC-derived cell therapies alongside rehabilitation, pharmacological interventions, or other regenerative medicine approaches, may enhance treatment outcomes. A deeper understanding of the molecular mechanisms underlying the regenerative capacity of iPSC-derived cells could also optimize their therapeutic benefits.

As the number of clinical trials increases, establishing appropriate patient selection criteria and outcome measures for evaluating the efficacy of iPSC-derived cell therapies in a clinically meaningful manner is essential. By addressing these challenges and focusing on innovative approaches, researchers can move closer to assessing the full potential of iPSC-derived cell therapies in treating spinal cord injuries and thereby improve patient outcomes and quality of life.

## 5. Challenges and the Road Ahead

While iPSCs hold immense potential for advancing SCI research and treatment, several challenges must be addressed before these therapies can be widely implemented in the clinic. We summarize the key challenges faced in the development and implementation of iPSC-based therapies for SCI, along with potential solutions to overcome these obstacles. Addressing these challenges will be crucial for realizing the full potential of iPSCs in SCI treatment and improving patient outcomes (Figure 2).

### 5.1. Safety Concerns

The generation and clinical application of iPSCs carry potential risks, including genetic mutations and epigenetic alterations, raising concerns about tumorigenesis, genomic instability, and immune-related adverse events [35]. Specifically, tumorigenicity remains a critical issue, as undifferentiated or incompletely differentiated iPSCs can form teratomas or other tumors after transplantation. Additionally, genomic instability resulting from reprogramming may lead to unintended cellular behaviors and increase the potential for malignant transformation. Immune responses also pose significant concerns, especially in allogeneic transplantation settings, where host immune rejection may occur despite HLA matching or immunosuppressive therapies. To mitigate these risks, significant efforts have been directed towards developing safer reprogramming methods that reduce genetic and epigenetic changes, including the use of non-integrative vectors, small molecules, and defined culture conditions [131,132]. Rigorous quality control processes, such as extensive genomic analysis, thorough characterization of differentiation protocols, and comprehensive preclinical testing, have been established to identify and eliminate potentially harmful cell populations before clinical use. Additionally, ongoing monitoring systems for patients receiving iPSC-based therapies are critical to promptly detect and address any adverse events, facilitating timely clinical intervention [133]. By carefully addressing these safety issues through improved techniques, rigorous quality controls, and vigilant patient monitoring, researchers and clinicians can significantly enhance the safety and clinical feasibility of iPSC-based therapies for treating SCI and other conditions.

### 5.2. Generation and Validation of Human iPSCs

Generating iPSCs from human somatic cells involves several established methods. Initial reprogramming approaches primarily utilized integrating viral vectors, such as retroviral or lentiviral delivery of reprogramming factors (e.g., OCT4, SOX2, KLF4, and c-MYC) [9]. However, concerns over insertional mutagenesis and genomic instability associated with these integrative vectors have led to the development of safer alternatives, including non-integrative viral methods, such as Sendai virus-based reprogramming and episomal plasmid vectors [134]. These approaches significantly reduce the risks of genomic alterations, enhancing the clinical applicability of iPSC-derived cells. Moreover, chemical reprogramming techniques utilizing small molecules and transcription factors have emerged, offering enhanced control over cellular reprogramming with minimized safety risks [135]. Regardless of the method used, the quality and pluripotency of iPSCs must be rigorously validated through a series of stringent tests. These include confirming the expression of pluripotency markers (e.g., OCT4, NANOG, and SOX2), assessing the differentiation capacity into all three germ layers (endoderm, mesoderm, and ectoderm) [9], evaluating genomic stability through karyotyping and whole-genome sequencing [136], and ensuring the absence of residual reprogramming factors and unwanted genetic alterations. Furthermore, careful selection of source somatic cells, typically fibroblasts from skin biopsies or peripheral blood cells, is essential to ensure efficient reprogramming and minimize variability [137]. Rigorous validation and characterization of iPSC lines are critical to ensure their safety, consistency, and functional potential prior to clinical translation.

### 5.3. Survival and Integration

For iPSC-based therapies to be effective, it is essential that transplanted cells not only survive but also integrate into the host spinal cord and establish functional connections with existing neurons [41]. To address this challenge, researchers are working on optimizing transplantation protocols, refining cell culture conditions, and investigating innovative strategies to enhance the survival, integration, and functionality of iPSC-derived cells [138]. Some approaches include modulating the host immune response to minimize rejection [139], using biomaterial scaffolds or hydrogels to provide structural support and promote cell attachment [140], preconditioning the cells prior to transplantation to increase their resilience [141], and employing growth factors or other signaling molecules to promote cell survival and differentiation [142]. By investigating these various strategies, researchers hope to enhance the success of iPSC-based therapies in promoting spinal cord repair and functional recovery following injury.

### 5.4. Overcoming the Inhibitory Environment

The injured spinal cord presents a highly inflammatory and inhibitory environment that can impede the effectiveness of cell-based therapies [143]. Identifying ways to modulate this environment and promote a more permissive state for cell survival, integration, and regeneration is crucial. Potential strategies to address this challenge include the use of anti-inflammatory agents to dampen the immune response [144,145], matrix-modifying enzymes to break down inhibitory extracellular matrix components [146], gene therapies to modulate the expression of key inhibitory factors [147], and combination therapies that target multiple inhibitory pathways simultaneously [148]. Moreover, researchers are exploring the use of biomaterials and tissue engineering approaches to create more conducive environments for iPSC-derived cells within the injured spinal cord [149,150]. By developing a deeper understanding of the complex interplay between iPSC-derived cells and the host environment, it may be possible to devise innovative strategies that overcome the inhibitory barriers and maximize the therapeutic potential of iPSC-based interventions for SCI.

### 5.5. Functional Recovery

Although some studies have demonstrated improvements in functional recovery following iPSC-based interventions in animal models, the extent of these improvements and their translation to human patients remain uncertain [151,152]. While standardized functional assessment tools such as electrophysiological recordings and hindlimb locomotor scoring systems (e.g., the BBB scale in rodents) are commonly used in preclinical models, their suitability for clinical translation is limited. In human SCI studies, assessments primarily rely on the International Standards for Neurological Classification of Spinal Cord Injury (ISNCSCI) and the American Spinal Injury Association (ASIA) Impairment Scale (AIS), which evaluate motor and sensory function recovery. However, variability in outcome measures across clinical studies remains a challenge. Advanced neuroimaging techniques, such as diffusion tensor imaging (DTI) and functional MRI (fMRI), offer promising avenues for monitoring axonal integrity and neural network recovery post-iPSC transplantation, though their standardization for SCI applications is still under investigation [153,154]. To address these challenges, it is essential to establish harmonized functional assessment protocols applicable across both preclinical and clinical studies. Additionally, integrating rehabilitation strategies alongside iPSC-based therapies and developing experimental models that better mimic human SCI pathophysiology will be crucial for accurately evaluating the efficacy of iPSC-based interventions. These refinements will facilitate the translation of preclinical findings into clinically relevant therapies, ultimately improving patient outcomes. 

### 5.6. Scalability and Reproducibility

A significant challenge in iPSC-based therapies is ensuring that the cells can be generated in sufficient quantities and with consistent quality for clinical applications [155,156]. This requires not only the development of robust, scalable protocols for iPSC production and differentiation but also the establishment of reliable quality control measures to assess cell identity, purity, and functionality [157]. Implementing advanced biomanufacturing techniques, such as automation and bioreactor-based culture systems [158], will be vital to address this challenge and enable large-scale, standardized production of iPSC-derived cells.

### 5.7. Personalized Treatment Strategies

One of the most significant promises of iPSC-based therapies is the potential for personalized treatment, as iPSCs can be derived from a patient’s own cells [159,160]. However, developing and implementing individualized therapies presents numerous challenges, including understanding the variability in patient responses, determining optimal cell sources, and tailoring cell differentiation and transplantation protocols to each patient’s specific needs. To address these challenges, researchers must continue to refine patient-specific iPSC generation and differentiation methods, develop robust biomarkers to predict individual patient responses, and establish standardized protocols for monitoring and evaluating patient outcomes following iPSC-based interventions.

As the field of iPSC research continues to evolve, addressing these challenges will be critical to realizing the full potential of these groundbreaking cells in the treatment of SCI. By overcoming these obstacles and further refining our understanding of iPSC biology and SCI pathology, researchers and clinicians may be able to develop innovative, effective therapies that can transform the lives of those affected by these debilitating injuries. The road ahead may be fraught with challenges, but the promise of iPSC-based therapies offers a beacon of hope for a brighter future for SCI patients.

## 6. Conclusions

iPSCs have ushered in a new era in the field of SCI research, illuminating the intricate cellular and molecular mechanisms that underlie injury and recovery. These remarkable cells have enabled the development of innovative therapeutic strategies and laid the groundwork for patient-specific treatments, bringing us one step closer to finding effective interventions for these life-changing injuries. While substantial progress has been made, numerous challenges still need to be addressed before iPSC-based therapies can be successfully implemented in clinical settings. Dedicated researchers are tirelessly working on improving reprogramming techniques, ensuring the safety and efficacy of transplanted cells, and developing strategies to counteract the inhibitory environment of the injured spinal cord. As our understanding of iPSC biology and its potential applications in SCI treatment continues to evolve, there is growing optimism that these pioneering stem cells will spearhead groundbreaking advancements in SCI research and therapy. By overcoming the hurdles that lie ahead, we can harness the full potential of iPSCs to create innovative, personalized treatments that substantially enhance the quality of life for individuals affected by SCI. The path may be arduous, but the potential rewards are immense, offering a beacon of hope and healing to millions of people around the world who are grappling with the devastating consequences of SCI.

## Figures and Tables

**Figure 1 biology-14-00314-f001:**
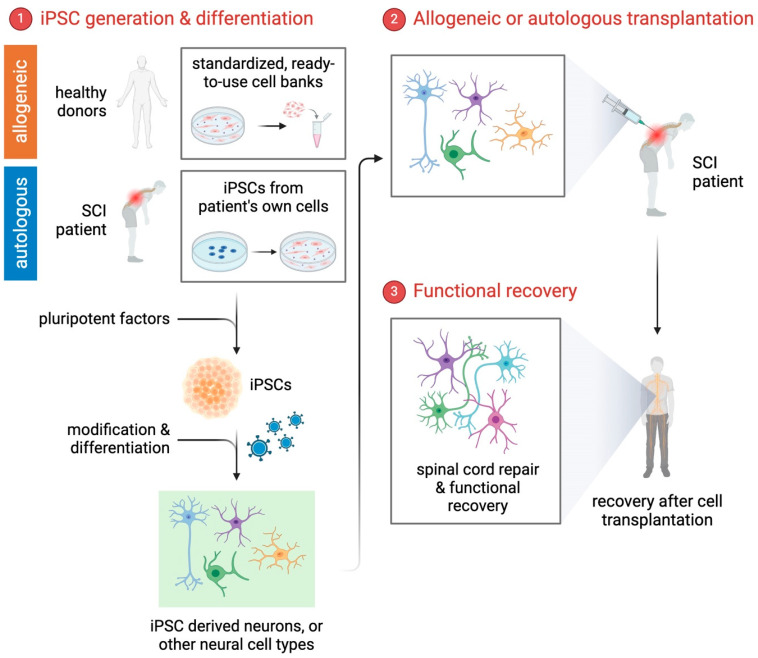
Schematic representation of the step-by-step differentiation process involved in iPSC-based therapy for SCI. Step 1 includes the generation of iPSCs from either allogeneic sources (healthy donors) or autologous sources (patient’s own cells). Allogeneic iPSCs offer standardized, readily available cells suitable for rapid clinical application, whereas autologous iPSCs minimize the risk of immune rejection but require longer preparation time. Step 2 involves differentiation of these iPSCs into various neural cell types relevant to SCI, including neurons, oligodendrocytes, astrocytes, and microglia. Step 3 encompasses the transplantation of these derived neural cells into the injured spinal cord, where they integrate into the host tissue, promoting tissue repair and functional recovery. This approach holds promise for developing personalized treatments for SCI patients.

**Figure 2 biology-14-00314-f002:**
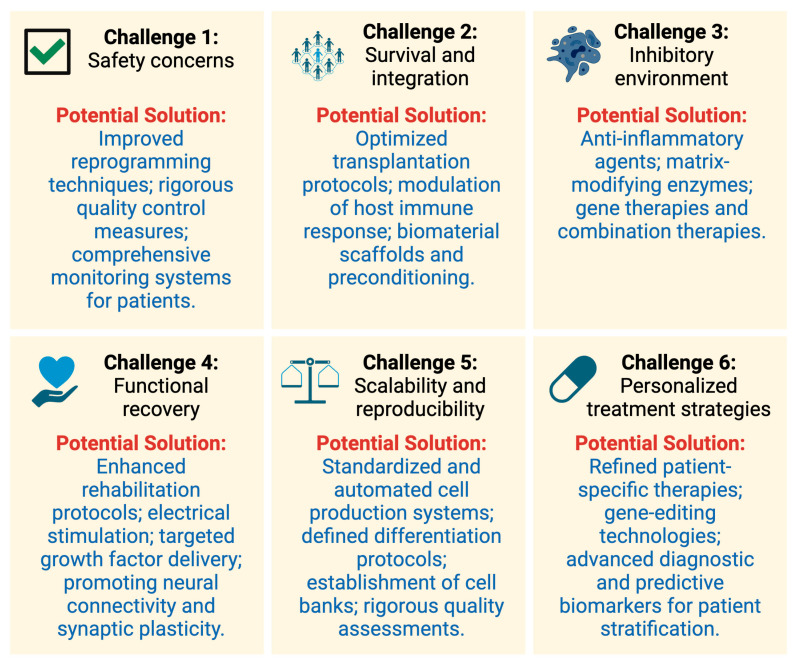
A schematic representation of the key challenges and potential solutions in the development and implementation of iPSC-based therapies for SCI. The figure features six interconnected challenge boxes addressing (1) safety concerns, (2) survival and integration, (3) the inhibitory environment, (4) functional recovery, (5) scalability and reproducibility, and (6) personalized approaches. Potential solutions specific to each challenge are outlined, highlighting critical strategies such as improved reprogramming methods, biomaterial scaffolds, optimized cell transplantation protocols, modulation of the host immune environment, enhanced rehabilitation protocols, standardized cell production systems, and personalized medicine strategies. The figure emphasizes the interconnected nature of these challenges and the necessity of integrated multidisciplinary efforts to fully realize the therapeutic potential of iPSCs for SCI treatment and improve patient outcomes.

**Table 1 biology-14-00314-t001:** Summary of iPSC-derived cell types, studies performed, and applications in the context of SCI.

Cell Type	Research Methods	Therapeutic Potential	References
iPSC-derived neural progenitor cells (NPCs)	Differentiation, transplantation, functional recovery assessment	Promote axonal regeneration, remyelination, synapse formation, modulate local immune response	[38,39,40,41,42]
iPSC-derived oligodendrocyte progenitor cells (OPCs)	Differentiation, transplantation, remyelination assessment	Promote remyelination, improve neuronal function, secrete neurotrophic factors, modulate inflammation	[49,50,51,52,53,54,55,56]
iPSC-derived astrocytes	Differentiation, transplantation, functional recovery assessment	Modulate inflammation, promote tissue repair, provide structural support to regenerating axons, regulate glial scar formation	[57,58,59,60,61,62,63,64,65,66,67]
iPSC-derived microglia	Differentiation, transplantation, functional recovery assessment	Modulate inflammation, regulate immune response, investigate signaling pathways	[68,69,70,71,72,73,74,75,76,77,78,79,80]
Combination therapies	Co-transplantation of different cell types, incorporation of other therapeutic strategies (biomaterial scaffolds, growth factors, electrical stimulation)	Enhance neuroprotection, promote cell survival, support functional recovery through combinatorial approaches	[81,82,83,84,85]

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
