# Peer review of "Stem Cell-Based Approaches for Spinal Cord Injury: The Promise of iPSCs"

_biology, 2025, doi:10.3390/biology14030314_

Round 1
Reviewer 1 Report
Comments and Suggestions for Authors
It is a descriptive and interesting article with slight observations.
1.- The author must adequately mention the meaning of all the acronyms mentioned and use them throughout the writing.
2.- The author is requested to expand slightly the information on adverse events when using iPSCs
3.- Give credit to Takahashi in collaboration with The Shinka Yamanaka Laboratory (lines 62-63).
4.- In figure 1, the word renair must be corrected and changed to repair.
Reviewer 2 Report
Comments and Suggestions for Authors
In this study, the authors explained the potential of iPSC for therapy spinal cord injury by discussing the strategies for driving it from neural cells to be applied in preclinical studies and personalized medicine. Authors have mentioned some related concerns.
However, they should address the issues:
- The introduction must be enriched with more data. Different classes of stem cells and the advantages of iPSCs to ESC and MSC.
- Authors should also explain spinal cord injury more and mention the events that cause it, like traumatic injury.
- Authors should explain the effects of nich on iPSC.
- Authors should mention the concerns of different media needed for inducing PSC.
- Please categorize the different biomaterials and immunomodulatory ones and their roles in inducing PSC and related mechanisms. And also the mechanism of GF.
- It is important to mention the potential of iPSC exosome iPSC for the therapy of spinal cord injury.
- There are not enough examples and comparisons for studies regarding the personalized medicine approach of iPSC usage.
- More evidence should be added for explanation.
Best
Comments on the Quality of English LanguageMinor English editing is needed.
Reviewer 3 Report
Comments and Suggestions for Authors
The review covers modern trends in the use of Induced pluripotent stem cells in Spinal cord injury. Fundamental and applied aspects of this treatment and diagnostic technique are analyzed. The review is quite complete and covers all the main aspects of this issue. However, there are a number of shortcomings in the format of presentation of the material, which, in my opinion, do not allow publishing this review in the presented form.
The main drawback is too superficial a description of the provided main areas of application of these cells. Each paragraph from the sections provides only general data on certain aspects of cell therapy. In my opinion, firstly, it is necessary to indicate how this or that direction of differentiation of pluripotent stem cells is implemented. Secondly, in each of the sections it is advisable to provide at least one example of restoration of functional indicators.
The disadvantages and difficulties in the use of this therapy are described very superficially. In particular, I consider it necessary to provide examples of complications such as malignancy, immune reactions, and to more fully substantiate their possible causes and methods of prevention.
In the section on clinical trials, it is advisable to provide data on the largest studies with reference to official sources, as well as intermediate results. Indicate whether there are completed studies and guidelines for this method of therapy.
It is advisable to create a separate section or describe in more detail in one of the existing sections the methods for obtaining these cells from humans.
The authors say that today there are no uniform approaches to assessing functional results. In this regard, it is advisable to describe which indicators there is agreement on, and which are debatable.
The above-described shortcomings can be corrected based on available sources. After their correction, the review can be published.
Reviewer 4 Report
Comments and Suggestions for Authors
This review discusses the potential use of iPSCs in treating spinal cord injury (SCI) and covers various aspects of stem cell therapy in SCI. The author has done a good job structuring the topic into multiple subtopics, including breakthroughs in iPSCs, the types of cells they can develop into, and the current status and challenges in the field. Below are some minor suggestions to improve the manuscript:
1. Introduction: Clearly state the gap or challenge in SCI treatment. Is SCI treatable? What are the current treatment options? What advantages do iPSCs offer? Addressing these questions directly will strengthen the introduction.
2. Figure 1: There are several typos in the figure. For example:
- SCI patien
- Spinal cord reair & Functioanl recovercy
Please carefully review and correct all typos.
3. Figure 1: iPSC Source: Why are iPSCs generated from healthy donors instead of using a patient’s own cells? What is the reasoning for choosing allogeneic over autologous transplantation? Clarifying this would improve the discussion.
4. Figure 2: The proposed solutions for challenges 4 and 5 appear to be the same as those for challenges 1 and 2. This is confusing—please review and clarify how challenges 4 and 5 are specifically addressed.
Round 2
Reviewer 2 Report
Comments and Suggestions for Authors.
Comments on the Quality of English LanguageMinor Engligh editing is needed.
Reviewer 3 Report
Comments and Suggestions for Authors
The authors have taken into account all comments and added all necessary information. The manuscript can be published, but the English needs to be checked.
Comments on the Quality of English LanguageThe authors have taken into account all comments and added all necessary information. The manuscript can be published, but the English needs to be checked.